# Maternal Undernutrition and Breast Milk Macronutrient Content Are Not Associated with Weight in Breastfed Infants at 1 and 3 Months after Delivery

**DOI:** 10.3390/ijerph16183315

**Published:** 2019-09-09

**Authors:** Takafumi Minato, Kyoko Nomura, Hitomi Asakura, Ayaka Aihara, Haruko Hiraike, Yuko Hino, Tsuyoshi Isojima, Hiroko Kodama

**Affiliations:** 1School of Medicine, Akita University, 1-1-1 Hondo, Akita city 010-8543, Japan; 2Department of Environmental Health Science and Public Health, Akita University Graduate School of Medicine, 1-1-1 Hondo, Akita city 010-8543, Japan; 3Department of Nutrition, Teikyo University Hospital, 2-11-1Kaga, Itabashi-ku, Tokyo 173-8605, Japan (H.A.) (A.A.); 4Department of Obstetrics and Gynaecology, Teikyo University School of Medicine, 2-11-1Kaga, Itabashi-ku, Tokyo 173-8605, Japan; 5Department of Pediatrics, Teikyo University School of Medicine, 2-11-1Kaga, Itabashi-ku, Tokyo 173-8605, Japan (Y.H.) (T.I.); 6Department of Health and Dietetics, Teikyo Heisei University, 2-51-4 Higashi Ikebukuro, Toshima-ku, Tokyo 170-8445, Japan

**Keywords:** breast milk, breastfed infant, infant weight, macronutrient content, undernutrition

## Abstract

This study examined whether maternal nutritional intake and breast milk macronutrient content influence the weight of breastfed infants. We investigated 129 healthy mothers with singleton babies born from July 2016 to December 2017 in a university hospital in Tokyo, Japan. Information was obtained by a self-administered food frequency questionnaire at 1 (valid response *n* = 92; mean age, 34 years) and 3 (*n* = 57) months after delivery. Breast milk was sampled at 1 and 3 months and the macronutrient contents were analyzed. The average pre-pregnancy body mass index and weight gain during pregnancy were 20.7 ± 2.6 kg/m^2^ and 9.6 ± 3.7 kg, respectively. At 1 month, average maternal calorie intake was 1993 ± 417 kcal/day, which was lower than the intake recommended by Japanese Dietary Reference Intakes for breastfeeding mothers. There were no significant differences with regard to maternal calorie and protein intake, and breast milk macronutrient content between breastfed infants with weight above and below the 25th percentile of its distribution at both 1 and 3 months. This study suggests that suboptimal calorie intake by breastfeeding mothers and breast milk macronutrient content were not associated with weight of their infants at 1 and 3 months after delivery.

## 1. Introduction

In 2017, the National Health and Nutrition Examination Survey in Japan showed that 21.7% of women in their 20s were underweight, defined as body mass index (BMI) < 18.5 kg/m^2^ [1]. National campaign to promote maternal and child health in 21st-century Japan (Healthy Parents and Children 21) [2] has raised concerns regarding the nutritional status of women of reproductive age in Japan because the proportion of underweight in both junior high and high school girls in Japan has increased to 20% [3]. Previous publications suggested that children born to such underweight women had an increased risk of infant mortality [4,5]. A recent database study [6] with more than 90,000 pregnant women compiled by the Japan Society of Obstetrics and Gynecology Successive Pregnancy Birth Registry System demonstrated that underweight women are more likely to bear small babies. There is accumulating evidence that low birth weight (birthweight < 2500 g) increases the risk of various health consequences in the offspring, even during their later lives [7,8]. This concept is known as the developmental origins of health and disease hypothesis, which posits that a poor nutritional environment during the fetal period adversely affects the fetus and increases risks of lifestyle-related diseases, such as cardiovascular disease or type 2 diabetes, after the child becomes an adult [8,9]. Although limited scientific evidence is available [10], a study demonstrated that maternal malnutrition, which results in prenatal exposure to excessive glucocorticoid levels, causes adverse metabolic programming, leading to hypertension in offspring [11]. Indeed, the prevalence of low birthweight infants in Japan has exceeded the average of 6.5% among Organization for Economic Co-operation and Development countries and reached >9% over the past 15 years [2,3,12]. Therefore, in 2001, the Japanese government set a goal to reduce this percentage, but no significant progress has been made to date.

Breastfeeding is a very important source of nutrition for infants born prematurely, all infants within 6 months after birth, and infants in developing countries where no artificial milk is available. The World Health Organization (WHO) strongly recommends exclusive breastfeeding for infants up to 6 months of age [13]. Although the nutritional composition of modern artificial milk is considered equivalent to that of breast milk [14], the benefits of breastfeeding are apparent for both mother and infant [15,16], including mother–infant bonding [15,16] and reduced risk of sudden infant death syndrome [15,17], certain allergic diseases such as asthma [15,16], obesity and diabetes [18], and breast and ovarian cancer [15,16]. Considering the central role of breastfeeding in infant growth, the nutritional status of mothers is tremendously important. However, there are scant scientific data on the growth of breastfed infants in relation to maternal nutrition status.

A study investigating the energy intake of lactating women and growth of infants indicated a slight positive association between maternal energy intake during lactation and infant growth at 2–3 years of age [19]. A study of 614 singleton births in the UK found that the carbohydrate concentration in breast milk showed a positive correlation with growth of the baby after 3 to 12 months [20]. Insufficient evidence is available regarding the relation between maternal nutrient intake and protein level in breast milk [21,22]. A positive correlation has been reported between dietary lipid intake and lipid levels in breast milk, but the sample sizes in these studies were very small [23,24]. Therefore, evidence is still required to clarify whether maternal nutrition affects breast milk components, and the extent of any influence on infant growth. This follow-up study 3 months after delivery was performed to examine maternal nutrition intake, breast milk macronutrient content, and weight of breastfed infants to understand the intake of energy and macronutrients by lactating mothers, the correlations between maternal nutrition and breast milk macronutrient content, and whether maternal nutrition and breast milk macronutrient content are associated with the weight of breastfed infants.

## 2. Materials and Methods

### 2.1. Participants

We recruited 129 consecutive healthy mothers who expected to give birth after 37 weeks of gestation of a single infant between July 2016 and December 2017 at Teikyo University Hospital, Department of Obstetrics and Gynecology, Tokyo, Japan (annual number of deliveries, 814 in 2017). We excluded (1) 28 women who did not provide sufficient responses to the food frequency questionnaire (FFQ); (2) 6 women who received medication for thyroid disease, high blood pressure, or diabetes who may have followed dietary restriction; and (3) 1 woman for whom the gestational week was not available. Therefore, 92 (71.3%) and 57 (44.2%) subjects (mean age 34.3 years) were included in the analyses at 1 and 3 months after delivery, respectively. The majority of mothers lived in, or temporarily returned to their parents’ home in, the neighborhood of the hospital. The self-administered FFQ for the first 1 month of the investigation was given to participants at the time of recruitment, with informed consent, and was sent to the homes of participants 1 month prior to the 3-month investigations. Responses were returned together with 30-mL breast milk samples by fast refrigerated delivery. To increase the response rate, we gave a $10 gift card coupon to each of our participants. We also distributed a leaflet to let our participants know that they could drop out of the follow-up survey any time they wanted to do so. This procedure was embedded in a research protocol that was approved by the Teikyo University School of Medicine Ethics Committee (Teikyo Lin No. 16-010-3).

### 2.2. Weights of Mothers and Infants and Maternal Nutrients at 1 and 3 Months after Delivery

We collected information on the self-reported weight and height of mothers before pregnancy, immediately before and after delivery, and at 1 and 3 months after delivery. For infants, information on weight, height, and head circumference were collected at birth and at 1 and 3 months after birth and were recorded in the Mother and Child Health Handbook distributed by the Japanese government. The nutritional intake of mothers was measured by FFQ, a self-administrated questionnaire based on 15 food and beverage groups with over 700 items from which intake of energy and 154 nutrients can be estimated using the Standardized Tables of Food Composition in Japan (2015 edition). We asked about the habitual consumption of listed foods within the past 2 months. Food portions on a one-week basis were queried as, “How much of this particular food (as illustrated) do you eat at any one time (i.e., morning)?” Portion size was specified for each food item using four standard sizes: “almost never”, “small (50% smaller)”, “medium (the standard amount)”, and “large (50% larger)”. The validity of this questionnaire has been reported on elsewhere [25]. However, because of our small sample size, to determine the validity of the FFQ, a 3-day food intake record was also measured in the first 50 consecutive mothers. Dietary records were analyzed by a dietitian, and interviews were performed as needed to obtain accurate information.

### 2.3. Definition of Breastfeeding Status

We collected information about breastfeeding at 1 and 3 months after delivery using a self-administered questionnaire. The breastfeeding status was classified as follows: (1) 100% breast milk; (2) 80% breast milk; (3) 50% breast milk and 50% artificial milk; (4) 80% artificial milk; (5) 100% artificial milk. Exclusive breastfeeding (EBF) has been defined by the WHO as the situation where “the infant has received only breast milk from his/her mother or a wet nurse, or expressed breast milk and no other liquids, or solids, with the exception of drops or syrups consisting of vitamins, minerals, supplements, or medicines [26].” We defined “exclusive/predominant” breastfeeding by combining “exclusive” (100%) and “predominant” (80%).

### 2.4. Breast Milk Collection and Measurement

Participants were advised to collect 30-mL breast milk samples in sterile conical tubes at 1 and 3 months after delivery. Immediately following extraction, samples were stored in a refrigerator at < 5 °C for a maximum period of 24 h prior to storage at −80 °C until thawing for analysis. Each frozen sample was initially heated at 40 °C in a thermostatically controlled bath. Samples of 1–3 mL were then homogenized to measure energy, carbohydrates, proteins, and lipids in the milk using a validated human milk analyzer (Miris holding, Uppsala, Sweden) [27], and calcium and phosphorus were analyzed using a multitype emission spectrometer (ICPE-9000; Shimadzu Corp., Kyoto, Japan). All samples were measured twice under the supervision of technicians well trained in use of the Miris human milk analyzer at Beanstalk Co. (Tokyo, Japan) and measured by technicians well-trained in use of the ICPE-9000 at Meiji Co. (Tokyo, Japan). The averages of the two measurements were used in the analyses.

### 2.5. Statistical Analysis

To assess the validity of the FFQ, we calculated the Pearson’s correlation coefficient between the 3-day dietary records and FFQ. To assess the generalizability of our findings, we conducted a test of significance using the *t*-test to investigate differences in the population mean between nutritional intakes of mothers in the present study (measured by the FFQ) and those of lactating women in the 2007–2011 National Health and Nutrition Examination Survey measured by semi-weighed dietary records. To determine whether maternal nutrition and breast milk macronutrient content differed according to the weight of breastfed infants, the weight of the infants was divided into binary categories: below the 25th percentile as the lower-normal weight group vs. above the 25th percentile defined as the normal weight group. For statistical comparison, we used the *t*-test for maternal nutrition and Wilcoxon’s rank-sum test for breast milk contents according to the distribution of the variable used. Finally, Spearman’s correlation coefficients were calculated to investigate correlations between mothers’ nutrition intake and breast milk contents at 1 and 3 months after delivery. The significance level was set at 5% (two tailed), and all analyses were conducted using SAS (version 9.4; SAS Institute, Cary, NC, USA).

## 3. Results

During the 3-month study period, not considering the level of exclusivity, almost all mothers included in the analysis responded that they breastfed their infant. At 1 and 3 months after delivery, 58/92 (63.7%) and 43/57 (75.4%) of mothers responded that they exclusively breastfed their infants (i.e., >80%), respectively.

Table 1 shows the weights of mothers and infants at 1 and 3 months. The mean ± standard deviation (SD) BMI of mothers before pregnancy and at 1 and 3 months were 20.7 ± 2.6 kg/m^2^, 21.7 ± 2.5 kg/m^2^ and 21.3 ± 2.8 kg/m^2^, respectively. The prevalence of underweight, defined as BMI < 18.5 kg/m^2^ was 8.8% among mothers in the present study. Weight gain during pregnancy (median (25%, 75%)) was 9.75 kg (7.5, 11.0) in underweight, 10 kg (8.0, 12.0) in normal weight, and 2.3 kg (2.0, 7.8) in overweight and obese mothers, respectively. These values fell within the range recommended by the Ministry of Health, Labor and Welfare [28]. The median (25%, 75%) infant weights for boys and girls were 3110 g (2872, 3304) and 3018 g (2764, 3208), respectively, at birth; 4420 g (4020, 4655) and 4010 g (3760, 4290), respectively, at 1 month; and 7054 g (6400, 7470) and 6486 g (5715, 6957), respectively, at 3 months. The median (25%, 75%) measurement date was 31 days (29, 35) for 1-month data, and 120 days (114, 125) for 3-month data.

Table 2 shows a comparison of maternal nutrition intake between the present study, where it was determined by FFQ, and the 2016 National Health and Nutrition Survey (30–49-year-old women), where it was measured by the semi-weighted record method [29]. Pearson’s correlation coefficients revealed positive correlations between FFQ and the 3-day dietary record only for energy intake (*r* = 0.277, *p* = 0.076) and protein (*r* = 0.351, *p* = 0.023). Based on this result, we compared protein and energy intakes only among women aged 30–49 years (i.e., this age corresponds to the mean age of our sample) in the 2016 Nutrition Survey; no significant differences were observed at 1 and 3 months after delivery. The energy intake was far below the recommended level of 2350–2650 kcal for lactating women in as defined by the Dietary Reference Intakes (2015 version) [30].

Table 3 shows maternal intake of energy and protein below and above the 25th percentile of infant weight at 1 and 3 months after delivery. At both 1 and 3 months, the mean level of maternal calorie and protein intakes did not differ between infants who were below and those who were above the 25th percentile in weight, even among mothers of breastfed infants. This lack of a significant relation was also confirmed when adjusted for infant age in days.

Table 4 shows the Spearman’s correlation coefficients of maternal nutrition intake and breast milk macronutrient content at 1 and 3 months after delivery. At 1 month, although not statistically significant, weak positive correlations (*p* < 0.1) were observed between energy intake and lipids (*r* = 0.180, *p* = 0.088) in breast milk, between protein intake and Carbohydrate (*r* = 0.175, *p* = 0.098) in breast milk, between Phosphorous intake and energy (*r* = 0.176, *p* = 0.095), and between phosphorous intake and lipids (*r* = 0.187, *p* = 0.078) in breast milk. At 3 months, however, no correlations were observed between maternal nutritional intake and breast milk macronutrient content.

Table 5 shows the results of Wilcoxon’s test of the median differences in breast milk macronutrient content according to infant weight below and that above the 25th percentile at 1 and 3 months after delivery. The median protein content in breast milk was higher (1.5 g vs. 1.4 g, respectively, *p* = 0.002), and that of carbohydrates was lower (7.4 g vs. 7.5 g, respectively, *p* = 0.014) in the group with lower infant weight gain than in the normal weight gain group. These associations were consistently observed when stratified with the “exclusive/dominant” breastfeeding groups (*p* = 0.020 and *p* = 0.048, respectively). However, there were no other significant relations between breast milk macronutrient content and infant weight at either 1 or 3 months.

## 4. Discussion

Given that maternal obesity has increased and attracted a great deal of attention in various Western countries, this is the first report to focus on maternal underweight, to investigate how maternal suboptimal nutrition affects breast milk content and infant weight for a short period of time after delivery. In this 3-month follow-up study after delivery, we found that maternal calorie and protein intakes fell below the recommended ranges proposed in the latest version of the 2015 Japanese Dietary Reference Intakes. In addition, we found that this suboptimal calorie intake among breastfeeding mothers, and their breast milk macronutrient content, were not associated with infant weight at 1 and 3 months after delivery. Our study also demonstrated that suboptimal calorie intake of mothers does not affect human milk macronutrient content. Among the important breast milk components, vitamin D and calcium are essential for growth and prevention of rickets in infants [31]. The negative result of our study, that maternal calcium intake did not correlate with calcium in breast milk, is in fact consistent with previous studies [32,33] that found no association between maternal dietary calcium intake and breast milk calcium concentrations, and supports interventions with dietary calcium or vitamin D that showed no effects on breast milk calcium concentrations [34,35].

Our sample assessed at 3 months had significantly lower maternal calorie and protein intake in comparison to the 2001 report from a nationally representative cohort of lactating mothers at 3 months after delivery (1979 ± 380 kcal and 69.8 ± 14.5 g, vs. 2167 ± 402 kcal and 80.7 ± 17.5 g, respectively, *p* < 0.001) [36]. Both those averages were lower than the 2015 Dietary Reference Intakes [30] for lactating women (i.e., 2350 kcal/day), but it should be noted that maternal nutrition status has worsened in our study. Despite such a significant difference in maternal nutrition between two points in time, the milk components in the first Japanese human milk survey in 1991 [37] and our present study are similar; the levels of energy, protein, lipid, carbohydrates, calcium, and phosphorus in human milk at 1 and 3 months after delivery were 69 g/dL and 66 g/dL, 1.5 g/dL and 1.2 g/dL, 3.7 g/dL and 3.6 g/dL, 7.4 g/dL and 7.3 g/dL, 28 mg/dL and 28 mg/dL, and 17 mg/dL and 14 mg/dL, respectively. This means that, assuming that the human milk components have been consistent since 1991, the nutritional value of breast milk may not change in parallel with maternal nutrition deterioration. Or rather, considering similar results in the breast milk components between 1991 and the present study and the discrepancy in maternal nutrition status between 2001 and the present study, the findings may suggest that the macronutrient content of breast milk is prioritized to be kept within the normal range under quasi-starvation conditions. If so, maternal nutrition may deteriorate constantly over the next few decades, and the macronutrient content of breast milk and consequent infant growth might eventually be influenced. A systematic review of the impact of maternal nutrition on breast milk composition covering 36 publications, including data on 1977 lactating women and their healthy full-term infants, concluded that the available information on this topic is scant and highly varied [38]. In this regard, future studies with larger sample sizes are warranted.

In our study, the majority of expectant mothers kept their weight under the upper limit of the weight gain recommendation by the Ministry [28]. In a previous Japanese study [39] among 1691 normal and underweight women, 54% of women wished to maintain their weight gain during pregnancy below the upper limit recommendation. The most common reason why women thought avoiding excessive weight gain was important was “for ease of delivery and/or her health and well-being”. Considering maternal weight might be determined by misguided perceptions of weight gain during pregnancy, mothers might also be concerned about the return to their pre-pregnancy weight. In fact, we found that the median of maternal recovery weight at 1 month was 2.0 kg/month, which is faster than the 0.8 kg/month recommended by the 2015 Japanese Dietary Reference Intakes [30]. Given that maternal nutrition status has become even worse in our study, as described above, this is an area of scientific research that needs to be further explored.

Several limitations should be considered in interpreting the results of this study. First, in addition to the small sample size, the present study was conducted at a university hospital where high-risk pregnant women are hospitalized for safe delivery. To minimize sampling bias, we recruited only healthy pregnant women with full-term singleton babies and further excluded women who had underlying illnesses that could affect dietary restriction. In our study, maternal nutritional intake was not significantly different from that of lactating mothers in the 2016 National Health and Nutrition Examination Survey. Furthermore, the average weights of mothers and infants were similar to those of women in their 30s in the 2016 National Survey and the 2010 Infants Physical Growth Survey [1,40]. Therefore, although our sample size was small, our results may be generalizable to some extent. Second, the average age of our sample was 34 years, which was slightly older than that of women who had first childbirth in the Tokyo area (32.3 years) according to 2018 demographic statistics. Thus, our results might have been influenced by age. Third, although the previous study showed that FFQ had some correlations with actual food consumption [25], we found little correlation between FFQ and the 3-day dietary records and thus we used only energy and protein to avoid confusion. Fourth, in our study, breast milk was sampled in a single spot collection, so one can argue that the results might have been confounded by variations. A previous systematic review demonstrated that breast milk composition was relatively stable between 2 and 12 weeks and maturation of milk was associated with reduced variability in protein content [41]. Hence our sample collected at 3 months might have been exempt from the influence of variation. Finally, we did not collect breast milk at 6 months because the prevalence of breastfeeding sharply declines when the majority of babies begin eating solid food and reduce breast milk intake. Although the prevalence of breastfeeding falls, it remains unknown whether suboptimal maternal nutrition status might affect infant growth after the study period of 3 months.

## 5. Conclusions

Despite the above-mentioned limitations, this study showed that the nutritional intake status of lactating mothers is lower than the dietary intake standard. However, such maternal suboptimal caloric intake was not correlated with breast milk macronutrient content, and neither maternal undernutrition status nor milk macronutrient content were associated with weight in breastfed infants.

## Figures and Tables

**Table 1 ijerph-16-03315-t001:** Weight of mothers and infants at 1 and 3 months.

**Mother**	***N***	**Body Weight (kg)**	**BMI (kg/m^2^)**	**Weight Difference**
Before pregnancy	92	53.0 ± 7.6	20.7 ± 2.6	-
Just before delivery	91	62.7 ± 8.4	24.5 ± 2.8	-
Weight gain during pregnancy (kg)	91	-	-	10.0 (11.7, 7.5) *
Immediately after delivery	81	58.1 ± 7.7	22.7 ± 2.6	-
1 month after delivery	89	55.4 ± 7.2	21.7 ± 2.5	-
Maternal recovery weight (kg)	78	-	-	2.0 (1.0, 3.4) *
3 months after delivery	55	54.5 ± 8.0	21.3 ± 2.8	-
Maternal recovery weight (kg)	47	-	-	4.0 (2.0, 5.7) *
**Infant**	***n***	**Body Weight (g)**	**Height (cm)**	**Weight Difference**
At birth	92	3042 (2794, 3300) *	50 (49, 51) *	-
1 month after birth	84	4197 (3802, 4490) *	53 (52, 55) *	-
Weight gain during 1 month (g)	84	-	-	1106 (904, 1363) *
3 months after birth	57	6690 (6147, 7395) *	63 (61, 64) *	-
Weight gain during 3 months (g)	57	-	-	3714 (2976, 4260) *

* Median (25%, 75%).

**Table 2 ijerph-16-03315-t002:** Comparison with the National Health and Nutrition survey and Japanese dietary reference.

	National Health and Nutrition Survey	Our Sample (by Food Frequency Questionnaire)		Japanese Dietary Reference (2015 Ver.)
	2016 Breastfeeding Women (*n* = 163)	1 Month (*n* = 92)	3 Months (*n* = 57)	30–49 Years Old Woman (Breastfeeding Women)
Energy (kcal)	1878 ± 540	1993 ± 417	1979 ± 380	2350
Protein (g)	67.7 ± 19.7	71.1 ± 14.4	69.8 ± 14.5	70
Protein (%Energy)		14.3 ± 1.5	14.2 ± 1.6	13–20

Based on two sample population mean difference.

**Table 3 ijerph-16-03315-t003:** Maternal energy and protein intakes according to infant weight above and below the 25th percentile at 1 and 3 months after birth.

	1 Month				3 Months			
	Total (*n* = 89)		Exclusive/Predominant Breastfed Infants (*n* = 57)		Total (*n* = 57)		Exclusive/Predominant Breastfed Infants (*n* = 43)	
	<25th Percentile (*n* = 23)	>25th Percentile (*n* = 66)	<25th Percentile (*n* = 14)	>25th Percentile (*n* = 42)	<25th Percentile (*n* = 14)	>25th Percentile (*n* = 43)	<25th Percentile (*n* = 12)	>25th Percentile (*n* = 31)
Energy (kcal)	2015 ± 449	1991 ± 417	2111 ± 525	2011 ± 409	1911 ± 340	2001 ± 394	1981 ± 310	2041 ± 370
Protein (g)	71.0 ± 12.1	71.2 ± 15.5	74.0 ± 12.8	70.9 ± 14.6	65.4 ± 11.3	71.2 ± 15.3	67.6 ± 10.4	72.8 ± 15.1
Energy producing nutrient balance								
Protein (%Energy)	14.2 ± 1.2	14.3 ± 1.6	14.2 ± 1.4	14.1 ± 1.7	13.8 ± 1.5	14.3 ± 1.6	13.7 ± 1.5	14.3 ± 1.7

Based on a *T*-test. Numbers do not add up to the total population (*n* = 92 for 1 month) due to missing data.

**Table 4 ijerph-16-03315-t004:** Spearman’s correlation coefficients of maternal nutritional intake and breast milk macronutrient content at 1 and 3 months after delivery.

**At 1 Month**	**Energy (kcal)**	**Protein (g)**	**Lipid (g)**	**Carbohydrate (g)**	**Calcium (mg)**	**Phosphorous (mg)**
Median (25%, 75%)	70.0 (61.0, 80.0)	1.4 (1.3, 1.6)	3.8 (2.7, 4.9)	7.4 (7.1, 7.6)	29.8 (26.7, 33.3)	16.6 (14.6, 18.3)
*n*	91	91	91	91	82	82
Mother’s nutritional intake						
Energy	0.171	−0.077	0.180 #	0.140	−0.075	0.097
Protein	0.165	−0.101	0.172	0.175 #	−0.070	0.040
Calcium	0.147	−0.122	0.159	0.117	−0.163	−0.023
Phosphorous	0.176 #	−0.127	0.187 #	0.165	−0.125	−0.002
**At 3 Months**	**Energy (kcal)**	**Protein (g)**	**Lipid (g)**	**Carbohydrate (g)**	**Calcium (mg)**	**Phosphorous (mg)**
Median (25%, 75%)	63.3 (53.0, 77.5)	1.1 (1.0, 1.2)	2.8 (2.0, 4.8)	7.6 (7.3, 7.7)	28.7 (25.9, 31.9)	13.4 (11.6, 14.4)
*n*	56	56	56	56	48	48
Mother’s nutritional intake						
Energy	0.067	0.055	0.053	0.086	0.030	0.240
Protein	−0.047	−0.013	−0.069	0.185	0.053	0.161
Calcium	−0.080	−0.004	−0.103	0.149	0.130	0.143
Phosphorous	−0.035	−0.023	−0.057	0.153	0.120	0.183

# *p* < 0.1.

**Table 5 ijerph-16-03315-t005:** Breast milk macronutrient content according to infant weight above and below the 25th percentile at 1 and 3 months after birth.

Breast Milk Macronutrient Content	1 Month				3 Months			
	Total (*n* = 88)		Exclusive/Predominant Breastfeeding (*n* = 56)		Total (*n* = 56)		Exclusive/Predominant Breastfeeding (*n* = 42)	
	<25th Percentile (*n* = 23)	>25th Percentile (*n* = 65)	<25th Percentile (*n* = 14)	>25th Percentile (*n* = 42)	<25th Percentile (*n* = 14)	>25th Percentile (*n* = 42)	<25th Percentile (*n* = 12)	>25th Percentile (*n* = 30)
Energy (kcal)	71 (64, 80)	70 (59, 81)	76 (67, 83)	72 (62, 83)	63 (52, 81)	63 (54, 78)	63 (51, 85)	60 (52, 75)
Protein (g)	1.5 (1.4, 1.8) *	1.4 (1.3, 1.6)	1.5 (1.4, 1.7) *	1.4 (1.3, 1.5)	1.2 (1.0, 1.2)	1.1 (1.0, 1.2)	1.2 (1.0, 1.2)	1.1 (1.0, 1.2)
Lipid (g)	3.9 (3.0, 4.9)	3.7 (2.7, 4.9)	4.3 (3.2, 5.1)	3.8 (2.8, 5.0)	2.6 (1.9, 4.9)	3.0 (2.1, 4.8)	3.0 (1.8, 5.3)	2.7 (1.7, 4.4)
Carbohydrate (g)	7.4 (7.0, 7.5) *	7.5 (7.2, 7.7)	7.4 (7.1, 7.5) *	7.6 (7.4, 7.7)	7.5 (7.5, 7.7)	7.6 (7.2, 7.7)	7.5 (7.5, 7.7)	7.6 (7.5, 7.7)
Calcium (mg)	30.9 (28.4, 32.5)	30.7 (27.7, 33.8)	30.9 (28.5, 32.5)	31.1 (28.0, 34.3)	30.0 (26.6, 32.5)	28.2 (24.9, 31.9)	30.1 (26.3, 34.3)	29.6 (27.1, 32.8)
Phosphorous (mg)	17.1 (15.2, 18.9)	17.2 (15.2, 18.8)	18.1 (17.0, 18.9)	17.2 (15.6, 18.8)	14.0 (12.8, 14.7)	12.8 (11.4, 14.4)	14.4 (12.9, 14.7)	13.8 (12.4, 14.4)

Wilcoxon’s rank sum test; * *p* < 0.05. Numbers do not add up to the total population (*n* = 92 for 1 month, *n* = 57 for 3 months) due to missing data.

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
