# Peer review of "Maternal Undernutrition and Breast Milk Macronutrient Content Are Not Associated with Weight in Breastfed Infants at 1 and 3 Months after Delivery"

_ijerph, 2019, doi:10.3390/ijerph16183315_

Round 1

Reviewer 1 Report

Dear authors,

Thank you for the opportunity to review your manuscript. I think this work is important and appreciate your research on this topic. I have a few comments to help improve the manuscript. 

(1) Rotate tables to fill the pages

(2) Table 1 should be written out in wide form rather than long form to remove repetition of row names

(3) There are a number of typographical errors in the manuscript. For example extra spaces in Table number counts ("( n=14)").

(4) Use the term "relation" when comparing two variables rather than "relationship"

(5) Was breastmilk sampled at 1 day or 1 month? This is unclear in some of the manuscript.

(6) If you state something is significant, include the p-value or confidence interval.

(7) Consider citing the following publications that relate to this work.

Declercq E, MacDorman M, Cabral H, Stotland N. Prepregnancy Body Mass Index and Infant Mortality in 38 U.S. States, 2012-2013. Obstet Gynecol. 2016 Feb;127(2):279-87.

Pace ND, Siega-Riz AM, Olshan AF, Chescheir NC, Cole SR, Desrosiers TA, Tinker SC, Hoyt AT, Canfield MA, Carmichael SL, Meyer RE; National Birth Defects Prevention Study. Survival of infants with spina bifida and the role of maternal prepregnancy body mass index. Birth Defects Res. 2019 Jul 19. doi: 10.1002/bdr2.1552. [Epub ahead of print]

The latter shows that maternal prepregnancy underweight and obesity were associated with higher infant mortality among infants born with spina bifida.

(8) Consider discussing the mechanism of interest as well as additional analyses that do not look at only correlations.

Best,

A.R.

Author Response

Re: Maternal undernutrition and breast milk macronutrients are not associated with weight in breastfed infants at 1 and 3 months after delivery

Response to the reviewer 1

We would like to express our deep gratitude for the useful comment. We explain our response to the comments of the reviewer below point by point.

#1 Rotate tables to fill the pages

We have updated Table 1 together with the response to the comment #2

#2 Table 1 should be written out in wide form rather than long form to remove repetition of row names

We have updated Table 1 by rotating the table and removing repetition of row names. In this revision, we have deleted head circumstance from Table 1 since they are not related to the main results of this manuscript.  

Table 1. Weight of mothers and infants at 1 and 3 months

Mother 

N

Body weight(kg)

BMI(kg/m2)

Weight difference

Before pregnancy

92

53.0 ± 7.6

20.7 ± 2.6

-

Just before delivery

91

62.7 ± 8.4

24.5 ± 2.8

-

Weight gain during pregnancy(kg)

91

-

-

10.0 (11.7, 7.5)*

Immediately after delivery

81

58.1 ± 7.7

22.7 ± 2.6

-

1 month after delivery

89

55.4 ± 7.2

21.7 ± 2.5

-

Maternal recovery weight(kg)

78

-

-

2.0 (1.0, 3.4)*

3 months after delivery

55

54.5 ± 8.0

21.3 ± 2.8

-

Maternal recovery weight(kg)

47

-

-

4.0 (2.0, 5.7)*

Infant

N

Body Weight(g)

Height(cm)

Weight difference

At birth

92

3042 (2794, 3300)*

50 (49, 51)*

-

1 month after birth

84

4197 (3802, 4490)*

53 (52, 55)*

-

Weight gain during 1 month(g)

84

-

-

1106 (904, 1363)*

3 months after birth

57

6690 (6147, 7395)*

63 (61, 64)*

-

Weight gain during 3 months(g)

57

-

-

3714 (2976, 4260)*

*Median(25%, 75%)

#3 There are a number of typographical errors in the manuscript. For example extra spaces in Table number counts ("( n=14)").

We thank the reviewer for pointing out this. We have erased those extra spaces in Tables.

#4 Use the term "relation" when comparing two variables rather than "relationship"

We have changed “relationship” to “relation” in the following places of the text.

P7, Line 177,

This lack of a significant relation was also confirmed when adjusted for infant age in days.

P8, Line 198

However, there were no other significant relations between breast milk macronutrients and infant weight at either 1 or 3 months.

#5 Was breastmilk sampled at 1 day or 1 month? This is unclear in some of the manuscript.

We collected breast milk sample on 1-day basis. Accordingly, we modified the following sentence on P 14, Line 278-279.

“Fourth, our breast milk sample was on a basis of 1-day measurement that one can argue the results might have been confounded with variations.”

And we have expanded the discussion about the variation of the measurement based on 1-day.

P14, Line 280-284

A previous systematic review demonstrated that breast milk composition changes throughout the lactation period [41]. Colostrum has higher protein and lower energy, fat, and lactose than does mature milk in both preterm and term births. Breast milk composition was relatively stable between 2 and 12 weeks. Maturation of milk was associated with reduced variability in protein content. Hence our sample collected at 3 months might have been exempted from the variation influence.

#6 If you state something is significant, include the p-value or confidence interval.

We found that the corresponding sentences that the reviewer suggested are “Our sample measured at 3 months had significantly lower maternal calorie and protein intake in comparison to the 2004 report from a nationally representative cohort of lactating mothers at 11 weeks after delivery (1979 ± 380 kcal and 69.8 ± 14.5 g, vs. 2167 ± 402 kcal and 80.7 ± 17.5 g, respectively, p < 0.001) [32]. Both those averages were lower than the 2015 Dietary Reference Intakes for lactating women (i.e., 2350 kcal/day). Despite of significant difference in maternal nutrition between two points in time, the milk components have been similar between the first Japanese human milk survey in 1991 and our present study; the components of energy, protein, lipid, carbohydrates, calcium, and phosphorus in human milk at 1 and 3 months after delivery were 69 g/dl and 66 g/dl, 1.5 g/dl and 1.2 g/dl, 3.7 g/dl and 3.6 g/dl, 7.4 g/dl and 7.3 g/dl, 28 mg/dl and 28 mg/dl, and 17 mg/dl and 14 mg/dl, respectively.”

The second “significant” sentence did not have any statistical values and this is why the reviewer suggested to include p-values or statistical numerical values. However, this “significant” refers to the previous “significant” in previous sentences. And the previous sentences had actually p-values. We believe that the reviewer would think the second “significant” would not relate to the previous “significant”. So in the revision, we have modified the sentence with the second “significant” as if it relates to the first significance sentence as follows.

Page 13, Line 263,

Despite of such a significant difference in maternal nutrition between two points in time, the milk components have been similar between the first Japanese human milk survey in 1991 and our present study;

Including “such a significant difference”, the reader would recognize that the significance relates to the previous significance written with p-value so redundant p-value can be refrained.

#7 Consider citing the following publications that relate to this work.

We have included the following references as ref # 4 and #5 in the revised manuscript.

Page 2, Line 43-44,

Literatures suggested such underweight women had the increased risk of infant mortality [4] or birth anomaly [5].

[4] Declercq E, MacDorman M, Cabral H, Stotland N. Prepregnancy Body Mass Index and Infant Mortality in 38 U.S. States, 2012-2013. Obstet Gynecol. 2016 Feb;127(2):279-87.

[5]Pace ND, Siega-Riz AM, Olshan AF, Chescheir NC, Cole SR, Desrosiers TA, Tinker SC, Hoyt AT, Canfield MA, Carmichael SL, Meyer RE; National Birth Defects Prevention Study. Survival of infants with spina bifida and the role of maternal prepregnancy body mass index. Birth Defects Res. 2019 Jul 19. doi: 10.1002/bdr2.1552. [Epub ahead of print]

#8 Consider discussing the mechanism of interest as well as additional analyses that do not look at only correlations.

We thank the reviewer for this helpful comment. Other reviewers also suggested us to expand the discussion to bring some merits for potential readers. We have updated discussion by touching on the mechanism of maternal undernutrition.

Discussion (Page 13 line 218-264)

Discussion

Given that maternal obesity is prevailed and has long been paid attention in Western countries, this is the first report to focus on maternal underweight and investigate how maternal suboptimal nutrition affects breast milk content and infant weight for a short period of time after delivery. In this 3-month follow-up study after delivery, we found that maternal calorie and protein intakes fell below the recommended ranges proposed in the latest version of the 2015 Japanese Dietary Reference Intakes. In addition, we found that this suboptimal calorie intake among breastfeeding mothers and their breast milk macronutrient content were not associated with infant weight at 1 and 3 months after delivery. Our study also demonstrated that suboptimal calorie intake of mothers do not affect human milk macronutrient content. Among the important breast milk contents, Vitamin D and Calcium are essential for the growth and the prevention of rickets in infants [31]. The negative result of our study that maternal Calcium intake did not correlate to Calcium in breast milk is in fact consistent with the previous studies [32,33] that found no association between maternal dietary calcium intake and breast milk calcium concentrations and supports the interventions with dietary calcium or vitamin D that showed no effects on breast milk calcium concentrations [34,35].

Our sample measured at 3 months had significantly lower maternal calorie and protein intake in comparison to the 2001 report from a nationally representative cohort of lactating mothers at 3 months after delivery (1979 ± 380 kcal and 69.8 ± 14.5 g, vs. 2167 ± 402 kcal and 80.7 ± 17.5 g, respectively, p < 0.001) [36]. Both those averages were lower than the 2015 Dietary Reference Intakes [30] for lactating women (i.e., 2350 kcal/day) but it should be noted that maternal nutrition status has been worsened in our study. Despite of such a significant difference in maternal nutrition between two points in time, the milk components have been similar between the first Japanese human milk survey in 1991 [37] and our present study; the components of energy, protein, lipid, Carbohydrates, Calcium, and phosphorus in human milk at 1 and 3 months after delivery were 69 g/dl and 66 g/dl, 1.5 g/dl and 1.2 g/dl, 3.7 g/dl and 3.6 g/dl, 7.4 g/dl and 7.3 g/dl, 28 mg/dl and 28 mg/dl, and 17 mg/dl and 14 mg/dl, respectively. This means that assuming that the human milk components are consistent since 1991, the nutrition of breast milk may not change in parallel with maternal nutrition deterioration. Or rather, considering similar results in the breast milk components between 1991 and the present study and the discrepancy in maternal nutrition status between 2001 and the present study, it may suggest that the macronutrient content of breast milk is prioritized to be kept within normal range under quasi-starvation conditions. If so, maternal nutrition will be constantly deteriorated over the next few decades, the macronutrient content of breast milk and consequent infant growth might be eventually influenced. A systematic review of the impact of maternal nutrition on breast milk composition covering 36 publications, including data on 1977 lactating women and their healthy full-term infants, concluded that the available information on this topic is scant and highly varied [38]. In this regard, future studies should be warranted with a larger sample size.

In our study, the majority of expecting mothers kept their weight under the upper limit of the weight gain recommendation by the Ministry [28]. A previous Japanese study [39] among 1,691 normal and underweight women, 54% women wished to maintain their weight gain during pregnancy below the upper limit recommendation and the most common reason why women thought avoiding excessive weight gain was important was "for ease of delivery and/or her health and well-being". Considering maternal weight might be determined by such wrong perception of weight gain during pregnancy, mothers might also be concerned about the return to their pre-pregnancy weight. In fact, we found that the median of maternal recovery weight at 1 month was 2.0 kg/month which is faster than 0.8kg/month recommended by the 2015 Japanese Dietary Reference Intakes [30]. Given that maternal nutrition status has become even worse in our study described above, this is the apparent area of scientific research that needs to be further explored.

Reviewer 2 Report

Overall this paper is well written and study design well described. Referencing and discussion could be improved, as indicated below.

Title: It would make more sense to rephrase the title as “maternal undernutrition and breast milk macronutrient content are not associated……” The same applies to the abstract and other places in the manuscript where this phrase is used.

Abstract:

Please state in the abstract that breast milk composition was analysed – it is not obvious from the way it is currently written.

Lines 30-33: the term “growth” usually refers to weight and length – as you did not measure length, suggest remove the word “growth” from line 32.

Line 31: “low calorie intake” : this would be better phrased as “suboptimal calorie intake”.

Introduction

Line 44: Please define what Low Birth Weight is.

Line 50: The references cited (7-9) are not appropriate and appear to have been chosen at random.  Reference 7 is about Selenium and neurodevelopment. Reference 8 is about vitamin A and asthma. Reference 9 is about depression and preterm infants. This article is not about any of those micronutrients or medical conditions, or about preterm infants. Recommend that alternative references are used, specifically those that focus on maternal undernutrition and DOHAD, and cite systematic reviews & meta analyses when available.

Line 60: “The WHO strongly recommends breastfeeding….” : this should say “exclusive breastfeeding”.

Materials and methods

Line 83:  It is unclear whether the participants were recruited retrospectively or prospectively – i.e. during pregnancy or after birth? Please state explicitly.

Line 98: Weight and height of mothers before pregnancy: was this measured or self-reported?

Line 101: FFQ: How was the list of foods for the FFQ derived? Was this based on a previously validated questionnaire or national study of commonly consumed foods? Please provide more detail of the FFQ– how many foods were included, did it specify portion sizes and over what time period did it record food intake (previous one month/one week)?

Line 107: Typo: “breeas”: should be “breast”.

Line 111:  I don’t agree with the definition of 80% breastfeeding as “exclusively” breastfeeding, as the WHO definition is very specific. 80% breastfeeding would be better described as “predominantly” breastfed. You can then combine “exclusive” and “predominantly” breastfed into one category of “exclusive/predominantly” for analysis, but the terminology is important.

Line 133: Please explain why the 25th centile has been chosen as a cut-off? Why not the 2nd  centile?  Is this standard practice in categorising growth in Japan?

Results

Table 5: Carbohydrate, calcium and phosphorous would be better with capital letters.

 Discussion

The discussion would benefit from some further discussion around the nutritional risks for the mother of suboptimal nutrition and breastfeeding (e.g. calcium and bone status).

Also seeing as this population is very different from the high rates of maternal obesity seen internationally, this is worth mentioning explaining.

Are there any specific cultural/socioeconomic issues relating to the Japanese diet which would contribute to the lower calorie intake observed (e.g. body image concerns to return to pre-pregnancy weight). Have any qualitative studies been conducted to help explain this?

Line 207: Full stop missing.

Author Response

Re: Maternal undernutrition and breast milk macronutrients are not associated with weight in breastfed infants at 1 and 3 months after delivery

Response to the reviewer 2

We would like to express our deep gratitude for the useful comment. We explain our response to the comments of the reviewer below point by point. We had professional English editing service in prior to the first submission.

#1 Title: It would make more sense to rephrase the title as “maternal undernutrition and breast milk macronutrient content are not associated……” The same applies to the abstract and other places in the manuscript where this phrase is used.

We have included “content” after “macronutrient” in title and some parts throughout the text.

#2 Abstract:

Please state in the abstract that breast milk composition was analysed – it is not obvious from the way it is currently written.

Abstract ,line 25-26, the following sentence was added.

The breast milk was sampled at 1 and 3 months and the macronutrient contents were analyzed.

#3 Lines 30-33: the term “growth” usually refers to weight and length – as you did not measure length, suggest remove the word “growth” from line 32.

We have deleted “growth” and change to “weight” throughout the text.

#4 Line 31: “low calorie intake” : this would be better phrased as “suboptimal calorie intake”.

We thank the reviewer for this helpful suggestion. Accordingly, we have changed the words to “suboptimal calorie intake” throughout the text.  

#5 Line 44: Please define what Low Birth Weight is.

We have included the definition as follows.

Line 47-48

There is accumulating evidence that low birth weight (birthweight <2500g) increases the risk of various health consequences in offspring, even during their later lives [7,8].

#6 Line 50: The references cited (7-9) are not appropriate and appear to have been chosen at random.  Reference 7 is about Selenium and neurodevelopment. Reference 8 is about vitamin A and asthma. Reference 9 is about depression and preterm infants. This article is not about any of those micronutrients or medical conditions, or about preterm infants. Recommend that alternative references are used, specifically those that focus on maternal undernutrition and DOHAD, and cite systematic reviews & meta analyses when available.

We thank the reviewer for this helpful suggestions. We have deleted the following reference.

[7] Amorós, R.; Murcia, M.; González, L.; Rebagliato, M.; Iñiguez; C.; Lopez-Espinosa, MJ.; Vioque, J.; Broberg, K.; Ballester, F.; Llop, S. Maternal selenium status and neuropsychological development in Spanish preschool children. Environ Res. 2018, 166, 215-222.

[8] Parr, CL.; Magnus, MC.; Karlstad, Hol.; Holvik, K.; Lund-Blix, NA.; Haugen, M.; Page, CM.; Nafstad, P.; Ueland, PM.; London, SJ.; Håberg, SE.; Nystad, W. Vitamin A and intake in pregnancy, infant supplementation, and asthma development: The Norwegian Mother and Child Cohort. Am J Clin Nutr. 2018, 107 (5), 789-798.

[9] Wang, H.; Zhou, H.; Zhang, Y.; Wang, Y.; Sun, J. Association of maternal depression with dietary intake, growth, and development of preterm infants: a cohort study in Beijing, China. Front Med. 2017 Nov 27. doi 10.1007 / s11684-017-0591-y.Front Pediatr. 2018 Oct 16; 6: 295. doi: 10.3389 / fped. 2018.00295. ECollection 2018

In order to be consistent with the sentences before and after the sentence citing these 3 references “A great deal of epidemiological research has been conducted regarding this issue around the world [7-9]“, we have decided to delete the sentence itself because the paragraph that contains the sentence has already cited several references to explain how much volume of evidence has been accumulated in this field and is still meaningful even if the references[7-9] were deleted.

#7 Line 60: “The WHO strongly recommends breastfeeding….” : this should say “exclusive breastfeeding”.

We have added “exclusive” in the sentence as follows.

Line 61-62

The World Health Organization (WHO) strongly recommends exclusive breastfeeding for infants up to 6 months of age [13].

#8 Line 83:  It is unclear whether the participants were recruited retrospectively or prospectively – i.e. during pregnancy or after birth? Please state explicitly.

The participants were recruited during pregnancy(prospectively). We are sorry for this confusion and accordingly have modified the sentence as follows.

Line 85-87

We recruited 129 consecutive healthy mothers who expected the birth after 37 weeks of gestation to a single infant between July 2016 and December 2017 at Teikyo University Hospital, Department of Obstetrics and Gynecology, Tokyo, Japan (annual number of deliveries, 814 in 2017).

#9 Line 98: Weight and height of mothers before pregnancy: was this measured or self-reported?

We collected self-reported weights of mothers and weights of infants recorded in the Mother and Child Health Handbook that are distributed to every single expecting mother by the Japanese government. Accordingly, we have changed the following sentences.

Line 103-107

We collected information on the self-reported weight and height of mothers before pregnancy, immediately before and after delivery, and at 1 and 3 months after delivery. For infants, information of the weight, height, and head circumference were collected at birth and at 1 and 3 months after birth that were recorded in the Mother and Child Health Handbook distributed by the Japanese government.

#10 Line 101: FFQ: How was the list of foods for the FFQ derived? Was this based on a previously validated questionnaire or national study of commonly consumed foods? Please provide more detail of the FFQ– how many foods were included, did it specify portion sizes and over what time period did it record food intake (previous one month/one week)?

The FFQ that we used in our study is the most frequently used FFQ among dieticians in Japan but not used in a national study like the Japan Public Health Center-based Prospective Study (JPHCPS). JPHCPS uses brief-type self-administered diet history questionnaire (BDHQ) instead, consisting of 167 food and beverage items and requiring only 15 mins to answer. The food portion investigated of BDHQ is exactly same as ours (described below) and the response frequency uses up to 10 frequency categories while ours uses the self-reported number of times. Previously a Japanese paper investigated the validity of FFQ that we used and BDHQ comparing with dietary records and found that both FFQ and BDHQ well correlates to dietary records except for salt consumption. 

For the reference of the reviewer, we have attached the abstract of the reference. 

Dokai., K.; Nishimura, S.; Miyatake., N. [Comparison of Diet Surveys among Female University Students Enrolled in a Training Course for Registered Dietitians.] Journal of Japanese Society of Shokuiku. 9(4). 365-368. 2015

https://www.jstage.jst.go.jp/article/shokuiku/9/4/9_365/_pdf/-char/en

The present cross-sectional study compares three diet surveys comprising dietary records (DR),the food frequency questionnaire (FFQ) and the brief-type self-administered diet history questionnaire (BDHQ) among 79 female university students enrolled in a training course for registered dietitians. The total energy intake determined using the DR was 1461±500 kcal. Fat and salt intake differed significantly between the DR and the FFQ, and vitamin A and salt intake differed significantly between the DR and the BDHQ. All parameters except salt intake correlated significantly between the DR and FFQ. Total energy, carbohydrate, vitamin B2 and vitamin C intake also correlated significantly between the DR and the BDHQ. These results might provide a useful reference for evaluating dietary surveys among female university students.

We have also included the following description about the validity and the details of the FFQ on page 3, line 107-117.

The nutritional intake of mothers was measured by FFQ, a self-administrated questionnaire based on 15 food and beverage groups with over 700 items from which intake of energy and 154 nutrients can be estimated using the Standardized Tables of Food Composition in Japan (2015 edition). We asked about the habitual consumption of listed foods within the past 2 months. Food portion based on a one-week basis was asked “how much do you eat this particular food (as illustrated) at one time (i.e., morning)?”. Portion size was specified for each food item using four standard sizes: “almost never, small (50% smaller), medium (the standard amount), and large (50% larger). The validity of this questionnaire was reported in elsewhere [25]. However, because of our small sample size, in order to determine the validity of the FFQ, a 3-day food intake record was also measured in the first 50 consecutive mothers. Dietary records were analyzed by a dietitian, and interviews were performed as needed to obtain accurate information.

#11 Line 107: Typo: “breeas”: should be “breast”.

We have corrected the word as suggested (line 114). We thank the reviewer for this mistypo.

#12 Line 111:  I don’t agree with the definition of 80% breastfeeding as “exclusively” breastfeeding, as the WHO definition is very specific. 80% breastfeeding would be better described as “predominantly” breastfed. You can then combine “exclusive” and “predominantly” breastfed into one category of “exclusive/predominantly” for analysis, but the terminology is important.

We deeply appreciate the reviewer’s comment on the terminology. We certainly agree with the reviewer suggestion and have changed as follows.

Line 125-126

We defined “exclusive/predominant” breastfeeding by combining “exclusive” as (1) 100% and “predominant” as (2) 80%.

# 13 Line 133: Please explain why the 25th centile has been chosen as a cut-off? Why not the 2nd  oercentile?  Is this standard practice in categorising growth in Japan?

We thank the reviewer for pointing out how we categorize the continuous variables. We usually categorize continuous variables divided by its median of distribution as the reviewer suggested. The reason why we used 25% as cut-off point to divide into binary is just an idea of sensitive analysis. That means, if there is any association between cause and outcome, the risk estimated would be stronger (or enhanced) in the categorization divided by 25 % (or 75%) of its distribution than in one divided by its median. It does not make any difference if we reanalyze data by using binary categorization based on its median. However, we believe that if there is any chance to detect small difference, categorization based on upper or lower quartile is worth trying. We hope that the reviewer could understand our point.

#14  Table 5: Carbohydrate, calcium and phosphorous would be better with capital letters.

We thank the reviewer for this important suggestion and accordingly have changed with capital letters throughout the text.

#15 The discussion would benefit from some further discussion around the nutritional risks for the mother of suboptimal nutrition and breastfeeding (e.g. calcium and bone status).

We thank the reviewer for very helpful suggestion and included the following paragraph in Discussion on page 13, Line 226-231.

Among the important breast milk contents, Vitamin D and Calcium are essential for the growth and the prevention of rickets in infants [31]. The negative result of our study that maternal Calcium intake did not correlate to Calcium in breast milk is in fact consistent with the previous studies [32,33] that found no association between maternal dietary calcium intake and breast milk calcium concentrations and supports the interventions with dietary calcium or vitamin D that showed no effects on breast milk calcium concentrations [34,35].

31 Dror, D.K.; Allen, L.H.Overview of Nutrients in Human Milk. Adv Nutr. 2018, 9(suppl_1),278S-294S.

32  Feeley, R.M.; Eitenmiller, R.R.; Jones, J.B.; Jr., Barnhart, H. Calcium, phosphorus, and magnesium contents of human milk during early lactation. J Pediatr Gastroenterol Nutr. 1983,2,262–7.

33 Vaughan, L.A.; Weber, C.W.; Kemberling, S.R. Longitudinal changes in the mineral content of human milk. Am J Clin Nutr. 1979,32,2301–6.

34  Nickkho-Amiry, M.; Prentice, A.; Ledi, F.; Laskey, M.A.; Das, G.; Berry, J.L.; Mughal, M.Z. Maternal vitamin D status and breast milk concentrations of calcium and phosphorus. Arch Dis Child. 2008,93,179.

35  Basile, L.A.; Taylor, S.N.; Wagner, C.L.; Horst, R.L.; Hollis, B.W. The effect of high-dose vitamin D supplementation on serum vitamin D levels and milk calcium concentration in lactating women and their infants. Breastfeed Med. 2006,1,27–35.

#16 Also seeing as this population is very different from the high rates of maternal obesity seen internationally, this is worth mentioning explaining.

We thank the reviewer for raising this important issue and accordingly have included the following sentence in the first sentence of Discussion on page 13, line 218-226

.

Given that maternal obesity is prevailed and has long been paid attention in Western countries, this is the first report to focus on maternal underweight and investigate how maternal suboptimal nutrition affects breast milk content and infant weight for a short period of time after delivery. In this 3-month follow-up study after delivery, we found that maternal calorie and protein intakes fell below the recommended ranges proposed in the latest version of the 2015 Japanese Dietary Reference Intakes. In addition, we found that this suboptimal calorie intake among breastfeeding mothers and their breast milk macronutrient content were not associated with infant weight at 1 and 3 months after delivery. Our study also demonstrated that suboptimal calorie intake of mothers do not affect human milk macronutrient content.

#17 Are there any specific cultural/socioeconomic issues relating to the Japanese diet which would contribute to the lower calorie intake observed (e.g. body image concerns to return to pre-pregnancy weight). Have any qualitative studies been conducted to help explain this?

Again we thank the reviewer for this public health concern. Yes, we have one very important literature about psychological reasons for mothers to keep underweight during pregnancy.

Accordingly we have included the following sentences in Discussion on page 13,Line 253-263 .

In our study, the majority of expecting mothers kept their weight under the upper limit of the weight gain recommendation by the Ministry [28]. A previous Japanese study [39] among 1,691 normal and underweight women, 54% women wished to maintain their weight gain during pregnancy below the upper limit recommendation and the most common reason why women thought avoiding excessive weight gain was important was "for ease of delivery and/or her health and well-being". Considering maternal weight might be determined by such wrong perception of weight gain during pregnancy, mothers might also be concerned about the return to their pre-pregnancy weight. In fact, we found that the median of maternal recovery weight at 1 month was 2.0 kg/month which is faster than 0.8kg/month recommended by the 2015 Japanese Dietary Reference Intakes [30]. Given that maternal nutrition status has become even worse in our study described above, this is the apparent area of scientific research that needs to be further explored.

Ref 39 Ogawa, K.; Morisaki, N.; Sago, H.; Fujiwara, T.; Horikawa, R. Association between women's perceived ideal gestational weight gain during pregnancy and pregnancy outcomes. Sci Rep. 2018, 8(1),11574.

Line 207: Full stop missing.

We have inserted a period (Line 220). We thank the reviewer for checking the details.

Reviewer 3 Report

This study assessed whether maternal nutritional intake and breast milk macronutrients influence the weight of breastfed infants. The paper was well-written, however, the following points are to be addressed to justify the findings.

Recruited participants were only one university hospital in Tokyo. I assume health mothers who could give birth at such a hospital would be wealthy, well-educated or with severe diseases(?). You mentioned it in the limitation section, but still, mean age 34 would be relatively high. It may be biased. Could it possible to explain that bias and describe the detail both mothers and the hospital in Japan? Although generalisability is one of the main issues of this paper, the results of table 2 were not in the main flow of this paper. You may want to delete or put it in the method section. The results of table 5 are clinically important? (e.g., 1.5g vs. 1.4g or 7.4g vs. 7.5g)

Author Response

Re: Maternal undernutrition and breast milk macronutrients are not associated with weight in breastfed infants at 1 and 3 months after delivery

Response to the reviewer 3

We would like to express our deep gratitude for the useful comment. We explain our response to the comments of the reviewer below point by point.

#1 Recruited participants were only one university hospital in Tokyo. I assume health mothers who could give birth at such a hospital would be wealthy, well-educated or with severe diseases(?).

We agree with the reviewer in that our subjects might have had underlying illness. This is a reason why we excluded mothers who had thyroid disease, hypertension, diabetes because these illnesses might affect their eating habits like salt restriction. These exclusion criteria were described in method on page 2, Line 88-91,

.

We excluded 1) 28 women who did not provide sufficient responses to the food frequency questionnaire (FFQ); 2) 6 women who received medication for thyroid disease, high blood pressure, or diabetes who may have followed dietary restriction; and 3) 1 woman for whom the gestational week was not available.

#2 You mentioned it in the limitation section, but still, mean age 34 would be relatively high. It may be biased. Could it possible to explain that bias and describe the detail both mothers and the hospital in Japan?

We thank the reviewer and thus included the following sentences in study limitation on page 14, Line 273-276.

Second, the average age of our sample was 34 years which was slightly older than women who had 1st child birth in Tokyo area (i.e., 32.3 years) according to 2018 Demographic Statistics. Thus, our results might have been influenced by age.

#3 Although generalisability is one of the main issues of this paper, the results of table 2 were not in the main flow of this paper. You may want to delete or put it in the method section.

Thank you for the reviewer for the comment. However, we believe Table 2 would be helpful for readers to visually understand the generalizability in terms of maternal nutrition status. We tentatively left Table 2 as it was but are happy to follow the further request.

#4 The results of table 5 are clinically important? (e.g., 1.5g vs. 1.4g or 7.4g vs. 7.5g)

We believe that the relations between breast milk macronutrient content and the weight of infant are clinically very important. Although this study failed any significance, we believe such a negative result may still bring new insight in this area of research.

Reviewer 4 Report

Thank you very much for submitting your article to this journal. I read your article with interest. I think it's well-written. I have provided a few minor comments below. Please confirm my comments.

Minor comments

1. Methods

Please add more detail about procedure of informed consent (e.g., how did researcher make consideration for mothers not to feel pressure to cooperate in this study, did the participant mothers get any reward or compensation for this research, etc.)

2. Discussion

1) Although the findings of this study showed that a low calorie intake for mothers does not affect human mild macronutrients nor the growth of breastfed infants, the authors need to be careful interpreting the findings since these may give underweight mothers a misunderstanding of their maternal nutritional status. Mothers who have a low-calorie intake will have an influence on the growth of children after the period of this study (one month and three months after birth). Thus, it would be better to add further explanations referring to the previous studies that were mentioned in the introduction section (i.e., the proportion of underweight children in both junior high and high school girls in Japan has increased to 20%, and those who are underweight are more likely to bear small babies; low birth weight increases the risk of various health consequences in offspring, etc).

2) This study may add new insight to maternal under-nutrition and breast milk macro-nutrition but it is not clear. Please add discussions about the findings in this study using references in the introduction section [20-25].

Author Response

Re: Maternal undernutrition and breast milk macronutrients are not associated with weight in breastfed infants at 1 and 3 months after delivery

Response to the reviewer 4

We would like to express our deep gratitude for the useful comment. We explain our response to the comments of the reviewer below point by point.

#1 Please add more detail about procedure of informed consent (e.g., how did researcher make consideration for mothers not to feel pressure to cooperate in this study, did the participant mothers get any reward or compensation for this research, etc.)

We thank the reviewer for the comment. Yes, we have given $10 gift card coupon to each of our participants. We also distributed a leaflet to let our participants know that they were able to drop out from the follow-up survey anytime they wanted to do so. This procedure was embedded in the research protocol that was approved by the ethical committee. We have included the following sentences to respond to the reviewer’s comment. 

On page 3, line 97-101

In order to increase the response rate, we had given $10 gift card coupon to each of our participants. We also distributed a leaflet to let our participants know that they were able to drop out from the follow-up survey anytime they wanted to do so. This procedure was embedded in the research protocol that was approved by the Teikyo University School of Medicine Ethics Committee (Teikyo Lin No. 16-010-3).

#2 Although the findings of this study showed that a low calorie intake for mothers does not affect human mild macronutrients nor the growth of breastfed infants, the authors need to be careful interpreting the findings since these may give underweight mothers a misunderstanding of their maternal nutritional status. Mothers who have a low-calorie intake will have an influence on the growth of children after the period of this study (one month and three months after birth). Thus, it would be better to add further explanations referring to the previous studies that were mentioned in the introduction section (i.e., the proportion of underweight children in both junior high and high school girls in Japan has increased to 20%, and those who are underweight are more likely to bear small babies; low birth weight increases the risk of various health consequences in offspring, etc).

We thank the reviewer for this important suggestion. We agree that we never know if suboptimal maternal nutrition status might affect infant growth after the study period of 3 months. Accordingly we have included the following sentence in the study limitation section.

On page 14, line 284-288.

 Finally, we did not collect breast milk at 6 months because the prevalence of breastfeeding sharply declines when the majority of babies begin eating solid food and reduced breast milk intake. Although the prevalence of breastfeeding falls, it remains unknown if suboptimal maternal nutrition status might affect infant growth after the study period of 3 months.

#3 This study may add new insight to maternal under-nutrition and breast milk macro-nutrition but it is not clear. Please add discussions about the findings in this study using references in the introduction section [20-25].

We thank the reviewer for such a helpful suggestion to expand the discussion in order to make this manuscript more meaningful. We have actually similar comments from other reviewers and updated Discussion by touching on the mechanism of maternal low intake of energy and the strength of this study.

On page 13, Line 217-264

. Discussion

Given that maternal obesity is prevailed and has long been paid attention in Western countries, this is the first report to focus on maternal underweight and investigate how maternal suboptimal nutrition affects breast milk content and infant weight for a short period of time after delivery. In this 3-month follow-up study after delivery, we found that maternal calorie and protein intakes fell below the recommended ranges proposed in the latest version of the 2015 Japanese Dietary Reference Intakes. In addition, we found that this suboptimal calorie intake among breastfeeding mothers and their breast milk macronutrient content were not associated with infant weight at 1 and 3 months after delivery. Our study also demonstrated that suboptimal calorie intake of mothers do not affect human milk macronutrient content. Among the important breast milk contents, Vitamin D and Calcium are essential for the growth and the prevention of rickets in infants [31]. The negative result of our study that maternal Calcium intake did not correlate to Calcium in breast milk is in fact consistent with the previous studies [32,33] that found no association between maternal dietary calcium intake and breast milk calcium concentrations and supports the interventions with dietary calcium or vitamin D that showed no effects on breast milk calcium concentrations [34,35].

Our sample measured at 3 months had significantly lower maternal calorie and protein intake in comparison to the 2001 report from a nationally representative cohort of lactating mothers at 3 months after delivery (1979 ± 380 kcal and 69.8 ± 14.5 g, vs. 2167 ± 402 kcal and 80.7 ± 17.5 g, respectively, p < 0.001) [36]. Both those averages were lower than the 2015 Dietary Reference Intakes [30] for lactating women (i.e., 2350 kcal/day) but it should be noted that maternal nutrition status has been worsened in our study. Despite of such a significant difference in maternal nutrition between two points in time, the milk components have been similar between the first Japanese human milk survey in 1991 [37] and our present study; the components of energy, protein, lipid, Carbohydrates, Calcium, and phosphorus in human milk at 1 and 3 months after delivery were 69 g/dl and 66 g/dl, 1.5 g/dl and 1.2 g/dl, 3.7 g/dl and 3.6 g/dl, 7.4 g/dl and 7.3 g/dl, 28 mg/dl and 28 mg/dl, and 17 mg/dl and 14 mg/dl, respectively. This means that assuming that the human milk components are consistent since 1991, the nutrition of breast milk may not change in parallel with maternal nutrition deterioration. Or rather, considering similar results in the breast milk components between 1991 and the present study and the discrepancy in maternal nutrition status between 2001 and the present study, it may suggest that the macronutrient content of breast milk is prioritized to be kept within normal range under quasi-starvation conditions. If so, maternal nutrition will be constantly deteriorated over the next few decades, the macronutrient content of breast milk and consequent infant growth might be eventually influenced. A systematic review of the impact of maternal nutrition on breast milk composition covering 36 publications, including data on 1977 lactating women and their healthy full-term infants, concluded that the available information on this topic is scant and highly varied [38]. In this regard, future studies should be warranted with a larger sample size.

In our study, the majority of expecting mothers kept their weight under the upper limit of the weight gain recommendation by the Ministry [28]. A previous Japanese study [39] among 1,691 normal and underweight women, 54% women wished to maintain their weight gain during pregnancy below the upper limit recommendation and the most common reason why women thought avoiding excessive weight gain was important was "for ease of delivery and/or her health and well-being". Considering maternal weight might be determined by such wrong perception of weight gain during pregnancy, mothers might also be concerned about the return to their pre-pregnancy weight. In fact, we found that the median of maternal recovery weight at 1 month was 2.0 kg/month which is faster than 0.8kg/month recommended by the 2015 Japanese Dietary Reference Intakes [30]. Given that maternal nutrition status has become even worse in our study described above, this is the apparent area of scientific research that needs to be further explored.

Round 2

Reviewer 1 Report

Dear Authors,

Thank you for your positive response to my feedback. Please see my response to your response in red font below.

Re: Maternal undernutrition and breast milk macronutrients are not associated with weight in breastfed infants at 1 and 3 months after delivery

Response to the reviewer 1

We would like to express our deep gratitude for the useful comment. We explain our response to the comments of the reviewer below point by point.

#1 Rotate tables to fill the pages

We have updated Table 1 together with the response to the comment #2

#2 Table 1 should be written out in wide form rather than long form to remove repetition of row names

We have updated Table 1 by rotating the table and removing repetition of row names. In this revision, we have deleted head circumstance from Table 1 since they are not related to the main results of this manuscript.

Table 1. Weight of mothers and infants at 1 and 3 months

Mother 

N

Body weight(kg)

BMI(kg/m2)

Weight difference

Before pregnancy

92

53.0 ± 7.6

20.7 ± 2.6

-

Just before delivery

91

62.7 ± 8.4

24.5 ± 2.8

-

Weight gain during pregnancy(kg)

91

-

-

10.0 (11.7, 7.5)*

Immediately after delivery

81

58.1 ± 7.7

22.7 ± 2.6

-

1 month after delivery

89

55.4 ± 7.2

21.7 ± 2.5

-

Maternal recovery weight(kg)

78

-

-

2.0 (1.0, 3.4)*

3 months after delivery

55

54.5 ± 8.0

21.3 ± 2.8

-

Maternal recovery weight(kg)

47

-

-

4.0 (2.0, 5.7)*

Infant

N

Body Weight(g)

Height(cm)

Weight difference

At birth

92

3042 (2794, 3300)*

50 (49, 51)*

-

1 month after birth

84

4197 (3802, 4490)*

53 (52, 55)*

-

Weight gain during 1 month(g)

84

-

-

1106 (904, 1363)*

3 months after birth

57

6690 (6147, 7395)*

63 (61, 64)*

-

Weight gain during 3 months(g)

57

-

-

3714 (2976, 4260)*

*Median(25%, 75%)

#3 There are a number of typographical errors in the manuscript. For example extra spaces in Table number counts ("( n=14)").

We thank the reviewer for pointing out this. We have erased those extra spaces in Tables.

Make sure to remove other typographical errors. I did not list them all.

#4 Use the term "relation" when comparing two variables rather than "relationship"

We have changed “relationship” to “relation” in the following places of the text.

P7, Line 177,

This lack of a significant relation was also confirmed when adjusted for infant age in days.

P8, Line 198

However, there were no other significant relations between breast milk macronutrients and infant weight at either 1 or 3 months.

#5 Was breastmilk sampled at 1 day or 1 month? This is unclear in some of the manuscript.

We collected breast milk sample on 1-day basis. Accordingly, we modified the following sentence on P 14, Line 278-279.

“Fourth, our breast milk sample was on a basis of 1-day measurement that one can argue the results might have been confounded with variations.”

This still is unclear to me and I think English editing is needed. Is this just one sample on day 1 after birth or is there breastmilk sampling every day?

And we have expanded the discussion about the variation of the measurement based on 1-day.

P14, Line 280-284

A previous systematic review demonstrated that breast milk composition changes throughout the lactation period [41]. Colostrum has higher protein and lower energy, fat, and lactose than does mature milk in both preterm and term births. Breast milk composition was relatively stable between 2 and 12 weeks. Maturation of milk was associated with reduced variability in protein content. Hence our sample collected at 3 months might have been exempted from the variation influence.

#6 If you state something is significant, include the p-value or confidence interval.

We found that the corresponding sentences that the reviewer suggested are “Our sample measured at 3 months had significantly lower maternal calorie and protein intake in comparison to the 2004 report from a nationally representative cohort of lactating mothers at 11 weeks after delivery (1979 ± 380 kcal and 69.8 ± 14.5 g, vs. 2167 ± 402 kcal and 80.7 ± 17.5 g, respectively, p < 0.001) [32]. Both those averages were lower than the 2015 Dietary Reference Intakes for lactating women (i.e., 2350 kcal/day). Despite of significant difference in maternal nutrition between two points in time, the milk components have been similar between the first Japanese human milk survey in 1991 and our present study; the components of energy, protein, lipid, carbohydrates, calcium, and phosphorus in human milk at 1 and 3 months after delivery were 69 g/dl and 66 g/dl, 1.5 g/dl and 1.2 g/dl, 3.7 g/dl and 3.6 g/dl, 7.4 g/dl and 7.3 g/dl, 28 mg/dl and 28 mg/dl, and 17 mg/dl and 14 mg/dl, respectively.”

Remove "of".

The second “significant” sentence did not have any statistical values and this is why the reviewer suggested to include p-values or statistical numerical values. However, this “significant” refers to the previous “significant” in previous sentences. And the previous sentences had actually p-values. We believe that the reviewer would think the second “significant” would not relate to the previous “significant”. So in the revision, we have modified the sentence with the second “significant” as if it relates to the first significance sentence as follows.

Page 13, Line 263,

Despite of such a significant difference in maternal nutrition between two points in time, the milk components have been similar between the first Japanese human milk survey in 1991 and our present study;

Including “such a significant difference”, the reader would recognize that the significance relates to the previous significance written with p-value so redundant p-value can be refrained.

#7 Consider citing the following publications that relate to this work.

We have included the following references as ref # 4 and #5 in the revised manuscript.

Page 2, Line 43-44,

Literatures suggested such underweight mothers women had the increased risk of infant mortality [4] or birth anomaly [5].

[4] Declercq E, MacDorman M, Cabral H, Stotland N. Prepregnancy Body Mass Index and Infant Mortality in 38 U.S. States, 2012-2013. Obstet Gynecol. 2016 Feb;127(2):279-87.

[5]Pace ND, Siega-Riz AM, Olshan AF, Chescheir NC, Cole SR, Desrosiers TA, Tinker SC, Hoyt AT, Canfield MA, Carmichael SL, Meyer RE; National Birth Defects Prevention Study. Survival of infants with spina bifida and the role of maternal prepregnancy body mass index. Birth Defects Res. 2019 Jul 19. doi: 10.1002/bdr2.1552. [Epub ahead of print]

See edit above. Reference [5] is about infant mortality by maternal BMI among infants with a birth defect not the risk of a birth defect so you can remove the "or birth anomaly" statement.

#8 Consider discussing the mechanism of interest as well as additional analyses that do not look at only correlations.

We thank the reviewer for this helpful comment. Other reviewers also suggested us to expand the discussion to bring some merits for potential readers. We have updated discussion by touching on the mechanism of maternal undernutrition.

What about analyses with more depth (multivariable regression, etc.)?

Discussion (Page 13 line 218-264)

Discussion

Given that maternal obesity is prevailed and has long been paid attention in Western countries, this is the first report to focus on maternal underweight and investigate how maternal suboptimal nutrition affects breast milk content and infant weight for a short period of time after delivery. In this 3-month follow-up study after delivery, we found that maternal calorie and protein intakes fell below the recommended ranges proposed in the latest version of the 2015 Japanese Dietary Reference Intakes. In addition, we found that this suboptimal calorie intake among breastfeeding mothers and their breast milk macronutrient content were not associated with infant weight at 1 and 3 months after delivery. Our study also demonstrated that suboptimal calorie intake of mothers do not affect human milk macronutrient content. Among the important breast milk contents, Vitamin D and Calcium are essential for the growth and the prevention of rickets in infants [31]. The negative result of our study that maternal Calcium intake did not correlate to Calcium in breast milk is in fact consistent with the previous studies [32,33] that found no association between maternal dietary calcium intake and breast milk calcium concentrations and supports the interventions with dietary calcium or vitamin D that showed no effects on breast milk calcium concentrations [34,35].

Our sample measured at 3 months had significantly lower maternal calorie and protein intake in comparison to the 2001 report from a nationally representative cohort of lactating mothers at 3 months after delivery (1979 ± 380 kcal and 69.8 ± 14.5 g, vs. 2167 ± 402 kcal and 80.7 ± 17.5 g, respectively, p < 0.001) [36]. Both those averages were lower than the 2015 Dietary Reference Intakes [30] for lactating women (i.e., 2350 kcal/day) but it should be noted that maternal nutrition status has been worsened in our study. Despite of such a significant difference in maternal nutrition between two points in time, the milk components have been similar between the first Japanese human milk survey in 1991 [37] and our present study; the components of energy, protein, lipid, Carbohydrates, Calcium, and phosphorus in human milk at 1 and 3 months after delivery were 69 g/dl and 66 g/dl, 1.5 g/dl and 1.2 g/dl, 3.7 g/dl and 3.6 g/dl, 7.4 g/dl and 7.3 g/dl, 28 mg/dl and 28 mg/dl, and 17 mg/dl and 14 mg/dl, respectively. This means that assuming that the human milk components are consistent since 1991, the nutrition of breast milk may not change in parallel with maternal nutrition deterioration. Or rather, considering similar results in the breast milk components between 1991 and the present study and the discrepancy in maternal nutrition status between 2001 and the present study, it may suggest that the macronutrient content of breast milk is prioritized to be kept within normal range under quasi-starvation conditions. If so, maternal nutrition will be constantly deteriorated over the next few decades, the macronutrient content of breast milk and consequent infant growth might be eventually influenced. A systematic review of the impact of maternal nutrition on breast milk composition covering 36 publications, including data on 1977 lactating women and their healthy full-term infants, concluded that the available information on this topic is scant and highly varied [38]. In this regard, future studies should be warranted with a larger sample size.

In our study, the majority of expecting mothers kept their weight under the upper limit of the weight gain recommendation by the Ministry [28]. A previous Japanese study [39] among 1,691 normal and underweight women, 54% women wished to maintain their weight gain during pregnancy below the upper limit recommendation and the most common reason why women thought avoiding excessive weight gain was important was "for ease of delivery and/or her health and well-being". Considering maternal weight might be determined by such wrong perception of weight gain during pregnancy, mothers might also be concerned about the return to their pre-pregnancy weight. In fact, we found that the median of maternal recovery weight at 1 month was 2.0 kg/month which is faster than 0.8kg/month recommended by the 2015 Japanese Dietary Reference Intakes [30]. Given that maternal nutrition status has become even worse in our study described above, this is the apparent area of scientific research that needs to be further explored.

This manuscript need English editing including the new added sections.

Thank you for your edits to this manuscript.

Best,

A.R.

Author Response

Re: Maternal undernutrition and breast milk macronutrients are not associated with weight in breastfed infants at 1 and 3 months after delivery

Response to the reviewer

We would like to express our deep gratitude for his/her very helpful comment to make our manuscript more suitable for publication. We explain our response to the comments of the reviewer below point by point. The corrected sentence is underlined here. We also had the professional English editing service in this 2nd round of revision.

#1 Make sure to remove other typographical errors. I did not list them all.

We went carefully through the text to check typographical errors. Thank you very much.

#2 Was breastmilk sampled at 1 day or 1 month? This is unclear in some of the manuscript.

1st round of revision

→We edited the sentence “Fourth, our breast milk sample was on a basis of 1-day measurement that one can argue the results might have been confounded with variations.”

→The reviewer said “this still is unclear to me and I think English editing is needed. Is this just one sample on day 1 after birth or is there breastmilk sampling every day?”

In this 2nd round of revision, we have professional English editing service edited the sentence on page 14, L280-284 as below.

Fourth, in our study, breast milk was sampled in a single spot collection, so one can argue that the results might have been confounded by variations. A previous systematic review demonstrated that breast milk composition was relatively stable between 2 and 12 weeks and maturation of milk was associated with reduced variability in protein content [41]. Hence our sample collected at 3 months might have been exempt from the influence of variation.

#3 Remove "of"

Yes, we have removed “of” from the sentence below.

Despite of significant difference in maternal nutrition(1st round of revision)→Despite significant difference in maternal nutrition(On page 13, L239, 2nd round of revision)

#4 Reference [5] is about infant mortality by maternal BMI among infants with a birth defect not the risk of a birth defect so you can remove the "or birth anomaly" statement.

Thank you for the reviewer’s explanation. Accordingly, we have changed the sentence on page 2, Line 44.

Literatures suggested such underweight mothers had the increased risk of infant mortality [4, 5].

[4] Declercq E, MacDorman M, Cabral H, Stotland N. Prepregnancy Body Mass Index and Infant Mortality in 38 U.S. States, 2012-2013. Obstet Gynecol. 2016 Feb;127(2):279-87.

[5]Pace ND, Siega-Riz AM, Olshan AF, Chescheir NC, Cole SR, Desrosiers TA, Tinker SC, Hoyt AT, Canfield MA, Carmichael SL, Meyer RE; National Birth Defects Prevention Study. Survival of infants with spina bifida and the role of maternal prepregnancy body mass index. Birth Defects Res. 2019 Jul 19. doi: 10.1002/bdr2.1552. [Epub ahead of print]

#5 Consider discussing the mechanism of interest as well as additional analyses that do not look at only correlations.

In the 1st round of revision, we explained as below.

→We thank the reviewer for this helpful comment. Other reviewers also suggested us to expand the discussion to bring some merits for potential readers. We have updated discussion by touching on the mechanism of maternal undernutrition.

Then, the comment from the reviewer was “What about analyses with more depth (multivariable regression, etc.)?”.

Because our sample size is relatively small, we are afraid that additional multivariable analyses may not change the current result (i.e., null association between suboptimal maternal nutrition, breast milk components, and infant weight gain.). Instead, we have just expanded discussion by referring to previous survey results on breast milk components and maternal nutrition.

On page 13, L234-255

Our sample assessed at 3 months had significantly lower maternal calorie and protein intake in comparison to the 2001 report from a nationally representative cohort of lactating mothers at 3 months after delivery (1,979 ± 380 kcal and 69.8 ± 14.5 g, vs. 2,167 ± 402 kcal and 80.7 ± 17.5 g, respectively, p < 0.001) [36]. Both those averages were lower than the 2015 Dietary Reference Intakes [30] for lactating women (i.e., 2,350 kcal/day), but it should be noted that maternal nutrition status has worsened in our study. Despite such a significant difference in maternal nutrition between two points in time, the milk components in the first Japanese human milk survey in 1991 [37] and our present study are similar; the levels of energy, protein, lipid, Carbohydrates, Calcium, and phosphorus in human milk at 1 and 3 months after delivery were 69 g/dl and 66 g/dl, 1.5 g/dl and 1.2 g/dl, 3.7 g/dl and 3.6 g/dl, 7.4 g/dl and 7.3 g/dl, 28 mg/dl and 28 mg/dl, and 17 mg/dl and 14 mg/dl, respectively. This means that, assuming that the human milk components have been consistent since 1991, the nutritional value of breast milk may not change in parallel with maternal nutrition deterioration. Or rather, considering similar results in the breast milk components between 1991 and the present study and the discrepancy in maternal nutrition status between 2001 and the present study, the findings may suggest that the macronutrient content of breast milk is prioritized to be kept within the normal range under quasi-starvation conditions. If so, maternal nutrition may deteriorate constantly over the next few decades, and the macronutrient content of breast milk and consequent infant growth might eventually be influenced. A systematic review of the impact of maternal nutrition on breast milk composition covering 36 publications, including data on 1,977 lactating women and their healthy full-term infants, concluded that the available information on this topic is scant and highly varied [38]. In this regard, future studies should be warranted with larger sample sizes.

#6 This manuscript need English editing including the new added sections.

In this 2nd round of revision, we had English editing service as the reviewer requested.